# Margin Adaptive DPO: Leveraging Reward Model for Granular Control in Preference Optimization

## Abstract

Direct Preference Optimization (DPO) has emerged as a simple and effective method for aligning large language models. However, its reliance on a fixed temperature parameter leads to suboptimal training on diverse preference data, causing overfitting on easy examples and under-learning from informative ones. Recent methods have emerged to counter this. While Identity Preference Optimization (IPO) addresses general overfitting, its uniform regularization can be overly conservative. The more targeted approach of $\beta$-DPO suffers from its own limitations: its batch-level adaptation applies a single, compromised temperature to mixed-margin pairs, its linear update rule can produce unstable negative $\beta$ values, and its filtering mechanism discards potentially useful training signals.

In this work, we introduce Margin-Adaptive Direct Preference Optimization (MADPO), a method that provides a stable, data-preserving, and instance-level solution. MADPO employs a practical two-step approach: it first trains a reward model to estimate preference margins and then uses these margins to apply a continuous, adaptive weight to the DPO loss for each individual training sample. This re-weighting scheme creates an effective target margin that is amplified for hard pairs and dampened for easy pairs, allowing for granular control over the learning signal.

We provide a comprehensive theoretical analysis, proving that MADPO has a well-behaved optimization landscape and is robust to reward model estimation errors. We validate our theory with experiments on a summarization task using human preference data. MADPO consistently outperforms strong baselines across a comprehensive sweep of decoding temperatures.

## 1 Introduction

Aligning Large Language Models (LLMs) with human preferences has become a cornerstone of modern AI, enabling models that are more helpful and harmless (Bai et al., 2022), follow complex instructions (Ouyang et al., 2022), and excel at sophisticated tasks (Ziegler et al., 2019). The dominant paradigm for this is learning from preference data, which has given rise to a range of powerful techniques. While early successes were driven by multi-stage methods like Reinforcement Learning from Human Feedback (RLHF), the field has recently seen the emergence of more direct and stable approaches, most notably Direct Preference Optimization (DPO) (Rafailov et al., 2023).

While DPO offers a more direct approach to preference alignment, its effectiveness is constrained by a critical factor: the joint influence of the temperature parameter, $\beta$, and the quality of the preference data. Seminal work in this area by Wu et al. (2024b) demonstrated that the optimal choice of $\beta$ is highly contingent on the reward margin of a given pair. Their analysis revealed that easy pairs with a large margin benefit from a high, conservative $\beta$ to prevent overfitting, whereas hard pairs with a subtle margin require a low, aggressive $\beta$ to ensure the learning signal is captured. The vanilla DPO framework, with its single fixed $\beta$ applied to all samples, is fundamentally unable to reconcile these competing requirements. This inherent tension has motivated recent work on adaptive regularization strategies, which aim to tailor the learning objective to the difficulty of each preference pair.

The challenge of adaptive regularization has led to several innovations that improve upon vanilla DPO. Identity Preference Optimization (IPO) (Azar et al., 2024), for instance, effectively mitigates the general overfitting issue by replacing the loss function with a squared-error objective. While not explicitly designed to resolve the tension between high- and low-margin data, its uniform target margin partially addresses the problem by regularizing easy pairs, though at the risk of being overly conservative on more informative examples. The most direct attempt to solve this is $\beta$-DPO (Wu et al., 2024b), which introduces adaptive, batch-level strategies. However, while demonstrating improved results, its mechanisms introduce significant new challenges. Its $\beta$-batch adaptation, for instance, is potentially unstable—producing a divergent negative $\beta$ for difficult data—and applies a single, compromised temperature to mixed-margin batches. Furthermore, its $\beta$-guided filtering approach can be data-inefficient, as it potentially discards very high and low margin samples that may still contain useful learning signals. These issues of potential instability, coarse granularity, and data inefficiency highlight the need for a solution that is not only instance-level and data-preserving, but also inherently stable.

In this paper, to address these challenges, we introduce Margin-Adaptive Direct Preference Optimization (MADPO), a method that precisely controls the DPO objective through a practical two-step process. First, we train a standard reward model to learn how strongly one response is preferred over another for each training example. Our approach then leverages this reward model to guide the DPO policy, which works by learning to match the preferences captured by the reward model. MADPO strategically modifies the strength of the preference signal from the reward model before showing it to the policy. For hard and informative pairs, it amplifies the signal to make the preference seem stronger, forcing the policy to learn more aggressively, achieving the same effect as a low $\beta$. Conversely, for easy and uninformative examples where the preference is already obvious, it dampens the signal to make the preference seem weaker, which provides a stabilizing, per-sample regularization, achieving the same effect as a high $\beta$. This strategic modification of the preference signal allows for granular, instance-level control, making the alignment process more data-efficient.

Our theoretical analysis validates the design of MADPO. We demonstrate that its instance-level weighting scheme successfully regularizes the learning objective for easy preference pairs while amplifying the signal for difficult ones, all while maintaining a well-behaved and stable optimization landscape with bounded gradients. Crucially, we also prove that this granular control is not brittle; our analysis provides a formal guarantee that the practical two-step algorithm is robust to the estimation errors inherent in reward modeling. These theoretical results establish MADPO as a principled and reliable method.

We validate our theoretical claims with a series of experiments on a real-world summarization task using the Reddit TL;DR dataset. Evaluated on a 7B parameter model, our results demonstrate that MADPO consistently outperforms strong baselines—including Vanilla DPO, IPO, and $\beta$-DPO across a comprehensive sweep of decoding temperatures. MADPO maintains a steady net win rate advantage, consistently outperforming the most competitive baseline, $\beta$-DPO, by approximately 5% across standard temperatures and expanding to an +11.3% gain under high-entropy sampling. Further analysis provides deeper insight into our method's mechanics: a detailed ablation study reveals that while the amplification mechanism drives the most substantial performance leaps, the explicit regularization component acts as a necessary safeguard for optimizing peak deterministic generation via greedy decoding. Finally, an empirical stress test with adversarial preference labels directly confirms our theoretical guarantees regarding gradient stability, demonstrating that MADPO's piecewise formulation actively prevents optimization collapse.

## 2 Preliminaries

The goal of preference alignment is to fine-tune a language model policy $\pi_\theta$, parameterized by $\theta$, using a dataset of human preferences $\mathcal{D} = \{(x, y_w, y_l)\}_{i=1}^N$. For each prompt $x$, $y_w$ is the response preferred over the response $y_l$. The alignment process is typically framed by modeling the probability of these preferences.

### 2.1 Reinforcement Learning from Human Feedback (RLHF)

The RLHF paradigm (Ouyang et al., 2022) aligns a policy in two main stages: reward modeling and policy optimization.

**1. Reward Modeling.** This stage aims to learn a reward model $r_\phi(x, y)$, parameterized by $\phi$, that reflects human preferences. The probability that $y_w$ is preferred to $y_l$ is modeled using the Bradley-Terry-Luce (BTL) framework (Bradley & Terry, 1952; Luce et al., 1959):

$$P(y_w \succ y_l | x; \phi^*) = \sigma(h_{\phi^*}(x, y_w, y_l))$$
$$= \sigma(r_{\phi^*}(x, y_w) - r_{\phi^*}(x, y_l)), \tag{1}$$

where $\sigma(\cdot)$ is the logistic function. The optimal reward model parameters, $\phi^*$, are found by maximizing the likelihood of the preference dataset, which corresponds to minimizing the following negative log-likelihood loss:

$$\mathcal{L}_{\mathrm{RM}}(r_\phi) = -\mathbb{E}_{(x, y_w, y_l) \sim \mathcal{D}} \left[ \log \sigma(h_\phi(x, y_w, y_l)) \right]. \tag{2}$$

**2. Policy Optimization.** In the second stage, the policy $\pi_\theta$ is then fine-tuned using the trained reward model $r_{\hat\phi}(x, y)$, where $\hat\phi$ is the empirical estimate of the optimal parameters $\phi^*$. The policy is optimized to maximize the expected reward while being regularized by a Kullback-Leibler (KL) divergence penalty against a reference policy $\pi_{\mathrm{ref}}$:

$$\max_\theta \quad \mathbb{E}_{x \sim \mathcal{D}_p, y \sim \pi_\theta(y|x)}[r_{\hat\phi}(x, y)] - \beta D_{\mathrm{KL}}(\pi_\theta(y|x) || \pi_{\mathrm{ref}}(y|x)).$$

Here, $\beta$ is a hyperparameter that controls the strength of the KL regularization, and $\mathcal{D}_p$ represents the dataset of prompts.

## 2.2 Direct Preference Optimization (DPO)

DPO (Rafailov et al., 2023) is an alternative that bypasses the explicit reward modeling and reinforcement learning stages. The key insight is that the optimal solution to the KL-regularized objective has a closed-form solution that connects the optimal policy $\pi_{\theta^*}$ to the optimal reward function $r_{\phi^*}$:

$$r_{\phi^*}(x, y) = \beta \log \frac{\pi_{\theta^*}(y|x)}{\pi_{\mathrm{ref}}(y|x)} + \beta \log Z(x), \tag{3}$$

where $Z(x)$ is the partition function that normalizes the distribution.

By substituting this mapping (Eq. 3) into the BTL preference model, the likelihood can be expressed directly in terms of the policy $\pi_\theta$. This allows for end-to-end optimization of the policy by minimizing a single negative log-likelihood loss:

$$\mathcal{L}_{\mathrm{DPO}}(\pi_\theta; \pi_{\mathrm{ref}}) = -\mathbb{E}_{(x, y_w, y_l) \sim \mathcal{D}} \left[ \log \sigma\left( \beta h_\theta(x, y_w, y_l) \right) \right]$$
$$= -\mathbb{E}_{(x, y_w, y_l) \sim \mathcal{D}} \left[ \log \sigma\left( \beta \log \frac{\pi_\theta(y_w|x)}{\pi_{\mathrm{ref}}(y_w|x)} - \beta \log \frac{\pi_\theta(y_l|x)}{\pi_{\mathrm{ref}}(y_l|x)} \right) \right].$$

For notational clarity, we define the following reward margin functions:

- The **explicit reward margin** from a reward model $r_\phi$:

$$h_\phi(x, y_w, y_l) = r_\phi(x, y_w) - r_\phi(x, y_l).$$

- The **implicit reward margin** from a policy $\pi_\theta$:

$$h_\theta(x, y_w, y_l) = \log \frac{\pi_\theta(y_w|x)}{\pi_{\mathrm{ref}}(y_w|x)} - \log \frac{\pi_\theta(y_l|x)}{\pi_{\mathrm{ref}}(y_l|x)}.$$

## 2.3 Limitations of Vanilla DPO

The relationship between the true reward function and the optimal policy is well-defined. From Eq. 3, we can see that the explicit reward margin is proportional to the implicit reward margin:

$$h_{\phi^*}(x, y_w, y_l) = \beta h_{\theta^*}(x, y_w, y_l).$$

Here, the inverse temperature $\beta$ acts as a global hyperparameter. A smaller $\beta$ encourages a larger difference in the policy's log-ratios for a given explicit reward margin, promoting more aggressive, confident updates. Conversely, a larger $\beta$ encourages more conservative updates.

However, recent work has revealed that using a single, static $\beta$ as a global hyperparameter is a significant limitation of the vanilla DPO framework, often leading to overfitting. As argued by Azar et al. (2024), this issue is particularly acute on finite datasets. If all annotators in a sample unanimously prefer one response, the empirical explicit reward margin becomes infinite. To match this, the DPO objective will push the learned log-policy difference, $h_{\hat{\theta}}$, to be arbitrarily large, causing the model to become overconfident and overfit to the winning response.

Complementing this finding, Wu et al. (2024b) suggest that a single $\beta$ is insufficient for handling the diverse quality of preference data. They find that easy pairs with a large explicit reward margin are best handled with a high $\beta$ (a more conservative update) to prevent overfitting. In contrast, hard pairs with a small, subtle margin require a low $\beta$ (a more aggressive update) to effectively learn the preference signal. This tension reveals the need for a more dynamic, instance-aware approach to regularization, which motivated the development of subsequent methods like IPO and $\beta$-DPO.

## 2.4   Identity Preference Optimization (IPO)

Identity Preference Optimization (IPO) (Azar et al., 2024) addresses the overfitting issue by replacing the log-likelihood objective with a squared-error loss,

$$\mathcal{L}_{\text{IPO}}(\pi_\theta, \pi_{ref}) = \mathbb{E}_{(x,y_w,y_l)\sim\mathcal{D}}\left[\left(h_\theta(x,y_w,y_l) - \frac{1}{2\beta}\right)^2\right].$$

The mechanism of IPO can be understood by analyzing the optimality condition of its loss function. The squared-error loss is minimized for any given sample when the implicit reward margin satisfies $\beta h_{\theta^*}(x,y_w,y_l) = 1/2$. Unlike DPO, which attempts to match the true explicit reward margin $h_{\phi^*}$, IPO effectively sets a single, uniform target margin of $1/2$ for every preference pair in the dataset. This has a dual effect: for hard pairs where the true explicit margin is small ($h_{\phi^*} < 1/2$), the higher target margin amplifies the learning signal. Conversely, for easy pairs where the true explicit margin is large ($h_{\phi^*} > 1/2$), the low target margin aggressively dampens the signal, which is the source of the regularization.

## 2.5   $\beta$-DPO

To address the limitations of a fixed temperature, $\beta$-DPO (Wu et al., 2024b) introduces adaptive strategies to modulate the learning process. It proposes two primary mechanisms:

**Batch-Level $\beta$ Adaptation ($\beta$-batch).**   This approach adapts the temperature $\beta$ for each training batch using a linear function of the batch's average implicit reward margin, $\bar{h}_\theta$. The batch-specific temperature, $\beta_{\text{batch}}$, is set as:

$$\beta_{\text{batch}} = \beta(1 + m \cdot (\bar{h}_\theta - h_0)), \quad \text{where} \quad \bar{h}_\theta = \frac{1}{|\text{batch}|} \sum_{(x,y_w,y_l)\in\text{batch}} h_\theta(x,y_w,y_l).$$

Here, $\beta$ is a base temperature, $m$ is a scaling factor between zero and one, and $h_0$ is a predetermined threshold. This allows batches with higher average implicit margins to be trained with a higher, more conservative $\beta_{\text{batch}}$.

**$\beta$ guided filtering.**   This approach modifies the training data by stochastically filtering each batch. For each sample $(x,y_w,y_l)$ in a batch, a score is computed using the probability density function of a Normal distribution, $\mathcal{N}(\cdot\,;h_0,\sigma^2)$, evaluated at the policy's current implicit margin, $h_\theta(x,y_w,y_l)$. The score is highest for samples whose margin is close to the mean $h_0$. A new, smaller batch is then formed by performing weighted random sampling without replacement from batch, where the probability of selecting a sample is proportional to its score. This method dynamically focuses training on examples of a target difficulty level, effectively down-sampling both overly easy and potentially noisy pairs.

# 3 Margin Adaptive Direct Preference Optimization (MADPO)

In this section, we introduce Margin Adaptive Direct Preference Optimization (MADPO), a method that enhances the DPO objective by adaptively re-weighting each training sample. The core idea is to modulate the loss based on the explicit reward margin, $h_\phi$, to amplify the learning signal from informative low-margin pairs while dampening it for easy, high-margin pairs to prevent overconfidence. This provides a more granular and flexible approach to regularization than IPO and $\beta$-DPO. We begin by detailing the central component of our method: a continuous, margin adaptive weight function. We then define the full MADPO loss function and describe the practical two-step algorithm for its optimization.

## 3.1 Margin Adaptive Weight

The central component of our MADPO method is the weight function which adaptively modulates the learning objective for each training sample based on its preference margin. This subsection provides a detailed exposition of this function's design, its hyperparameters, and the reasoning behind its piecewise structure. To simplify the notation, where the context is unambiguous, any function will be denoted by its symbol alone, suppressing the explicit dependence on its arguments for conciseness.

The core of our method is a coefficient function, $c : \mathbb{R} \to [c_{\min}, c_{\max}]$, which maps the explicit reward margin $h_\phi$ to a modulating scalar. This function is designed to be greater than 1 for low margins and less than 1 for high margins. It is defined as:

$$c(h_\phi) = c_{\min} + \frac{c_{\max} - c_{\min}}{1 + \left(\frac{c_{\max}-1}{1-c_{\min}}\right)\exp\left(\lambda(h_\phi - \tau)\right)},$$

Using this margin-dependent coefficient, we define a piecewise weighting function, $w(h_\phi)$, which selectively modifies the likelihood ratio.

$$w(h_\phi) = \begin{cases} \frac{\sigma(c(|h_\phi|) \cdot h_\phi)}{\sigma(h_\phi)} & \text{if } h_\phi > -\tau \\ 1 & \text{if } h_\phi \leq -\tau \end{cases} \tag{4}$$

The threshold $\tau > 0$ provides a concrete definition for what constitutes a high-margin or easy preference pair. This value can be chosen based on practitioner judgment or derived from the data. Specifically, any pair is classified as high-margin if its absolute explicit reward margin satisfies $|h_\phi| \geq \tau$, and as low-margin if its explicit margin satisfies $|h_\phi| < \tau$.

The parameter $c_{\max}$ acts as the amplifier for low-margin pairs. As the explicit margin $|h_\phi|$ approaches zero, the coefficient approaches $c_{\max}$. This forces the model to learn more aggressively from the most informative and subtle preferences. A higher value provides a stronger signal boost, but risks overfitting to noise in these difficult examples. By definition, this parameter must be greater than one to ensure amplification.

The parameter $c_{\min}$ acts as the dampener for high-margin pairs, setting a floor on the regularization. As the margin grows, the learned target gap is scaled down by a factor approaching $c_{\min}$. A value near 0 instructs the model to almost entirely ignore obvious preferences, preventing overconfidence. A value closer to 1 instructs the model to still learn from them, but with less intensity. By definition, this parameter must be between zero and one, $c_{\min} \in [0, 1)$, to ensure damping.

The parameter $\lambda$ controls the sharpness of the transition around the threshold $\tau$. A large $\lambda$ creates a steep, switch-like change, treating samples on either side of $\tau$ very differently. A small $\lambda$ creates a much more gradual and smooth transition from amplification to dampening.

While these intuitions provide strong starting points for setting the parameters, their optimal values are typically dataset-dependent. Therefore, the most rigorous approach is to perform a hyperparameter search, using a method like cross-validation on a held-out set of preference data to find the combination that yields the best empirical performance.

The piecewise nature of the weight function is a crucial design choice for ensuring training stability. While the re-weighting mechanism works as intended for most pairs, it can lead to undesirable behavior at the extremes of the margin distribution without this piecewise control.

We analyze the behavior of the core ratio $\sigma(c(h_{\phi^*}|) \cdot h_{\phi^*})/\sigma(h_{\phi^*})$, evaluated at the optimal reward model parameter $\phi^*$, in two distinct cases:

- **For large positive margins ($h_{\phi^*} \gg \tau$):** In this region, where $c \approx c_{\min}$, the weight function converges to a small, stable, positive value. This correctly applies a consistent penalty to all easy pairs, achieving the desired regularization effect.

- **For large negative margins ($h_{\phi^*} \ll -\tau$):** Without the piecewise cutoff, the weight function would explode. As $h_{\phi^*} \to -\infty$, the weight can be approximated by $w(h_{\phi^*}) \approx e^{(c_{\min}-1)h_{\phi^*}}$. Since $c_{\min} < 1$ and $h_{\phi^*}$ is negative, the exponent is positive and grows linearly with $|h_{\phi^*}|$. This growth in the weight would assign a massive, potentially infinite loss to these samples, leading to severe gradient instability.

The piecewise definition elegantly solves this problem. By setting $w(h_{\phi^*}) = 1$ for all $h_{\phi^*} \leq -\tau$, we cap this potential explosion. This ensures that samples with very large negative margins—which may be mislabeled or adversarial—are handled by the vanilla, stable DPO loss instead of causing the training to diverge. This design allows us to achieve our desired penalization for high positive margins without sacrificing the stability of the overall training process. The impact of this weighting on the final loss function is discussed in the following sections.

### 3.2 Loss Function and Optimization

Having defined the margin adaptive weight, $w(h_\phi)$, we now formally incorporate it into our final loss function. We then detail the practical, two-step procedure used to train a policy with this new objective.

**The MADPO Loss Function.** The MADPO loss for a single preference pair is the vanilla DPO log-likelihood, re-weighted by our margin-dependent weight function:

$$\mathcal{L}(\theta, \phi; x, y_w, y_l) = -w(h_\phi(x, y_w, y_l)) \log \sigma(\beta h_\theta(x, y_w, y_l)). \tag{5}$$

This loss depends on both the policy parameters, $\theta$, (through $h_\theta$) and the reward model parameters, $\phi$, (through $h_\phi$). To optimize this effectively, we employ a two-step approach.

**Step 1: Reward Model Estimation.** First, we obtain a high-quality estimate of the preference margins. This is achieved by training a standard reward model, $r_\phi$, on the preference dataset $\mathcal{D}$ to find the estimated parameters, $\hat{\phi}$. This step is identical to the reward modeling stage of traditional RLHF:

$$\hat{\phi} = \underset{\phi}{\arg\min} \; \mathcal{L}_{\mathrm{RM}}(r_\phi).$$

**Step 2: Margin Adaptive Policy Optimization.** Second, we treat the estimated reward parameters $\hat{\phi}$ as a fixed, ground-truth source of preference margins. These parameters are plugged into our MADPO loss function, which now becomes an objective solely for the policy. We then find the final policy parameters, $\hat{\theta}$, by minimizing the expectation of this loss over the dataset:

$$\hat{\theta} = \underset{\theta}{\arg\min} \; \mathcal{L}(\theta, \hat{\phi}) = \underset{\theta}{\arg\min} \; \mathbb{E}_{(x, y_w, y_l) \sim \mathcal{D}} \left[ \mathcal{L}(\theta, \hat{\phi}; x, y_w, y_l) \right].$$

This two-step process provides a stable and practical method for training a policy that is explicitly aware of the nuance and difficulty of the preference data it learns from.

## 4 Theoretical Analysis

In this section, we provide the theoretical justification for the MADPO algorithm. Our analysis proceeds in three parts. First, we establish that our method achieves its primary goal: in an idealized oracle setting, MADPO encourages the policy to be confident on informative, low-margin pairs while achieving the desired

regularization for high-margin pairs. Second, we prove a formal performance guarantee for our practical two-step algorithm, showing that the loss function is Lipschitz continuous and therefore robust to errors from the reward estimation stage. Finally, we demonstrate that the gradient and Hessian of the MADPO loss are scaled versions of their vanilla DPO counterparts, which shows that our method provides controllable training stability while retaining the well-behaved optimization landscape of DPO.

### 4.1 Oracle Characterization of Margin-Adaptive Regularization

In this section, we formally characterize the behavior of the MADPO loss function, showing how it achieves the dual goals of amplifying the learning signal for informative low-margin pairs and regularizing the objective for high-margin pairs. To isolate this mechanism, we conduct our analysis in an oracle setting, assuming access to the true optimal reward parameters, $\phi^*$.

We begin by characterizing the optimal implicit reward margin that minimizes the MADPO loss across the entire dataset.

**Proposition 4.1.** *Under the BTL model with an optimal reward model $r_{\phi^*}$, the optimal policy parameter $\theta^*$ that minimizes the MADPO loss $\mathcal{L}(\theta, \phi^*; x, y_w, y_l)$ satisfies the following for any preference pair $(x, y_w, y_l) \in \mathcal{D}$:*

$$\beta h_{\theta^*}(x, y_w, y_l) = c(|h_{\phi^*}(x, y_w, y_l)|) \cdot h_{\phi^*}(x, y_w, y_l).$$

This global optimality condition reveals that the policy is not optimized to simply match the true explicit reward margin, $h_{\phi^*}$, but rather a target scaled by the coefficient $c(|h_{\phi^*}|)$. This identity acts as the mathematical foundation for MADPO's amplification mechanism. For any subtle, low-margin pair ($|h_{\phi^*}| < \tau$), our method sets $c > 1$ by construction. Consequently, Proposition 4.1 dictates that the policy must learn an amplified preference target, aggressively increasing the separation in its log-ratios to a greater degree than what the explicit reward alone would suggest.

While Proposition 4.1 establishes the scaled target, we must also ensure that manipulating the coefficient $c$ serves as a reliable, predictable control mechanism – particularly when we want to suppress the learning signal.

**Proposition 4.2.** *Under the BTL model with an optimal reward model $r_{\phi^*}$, the optimal implicit reward $\beta h_{\theta^*}$ is monotonically increasing with respect to the coefficient $c \equiv c(|h_{\phi^*}|)$ for any preference pair $(x, y_w, y_l) \in \mathcal{D}$. Formally:*

$$\frac{\partial(\beta h_{\theta^*})}{\partial c} > 0.$$

This proposition provides the formal guarantee required for MADPO's regularization on high-margin data. It establishes that the learned implicit reward, $\beta h_{\theta^*}$, is strictly and monotonically controlled by $c$. For any easy preference pair ($|h_{\phi^*}| \geq \tau$), our method sets $c < 1$. The monotonic relationship guarantees that this reduced coefficient strictly shrinks the optimization target relative to the true explicit margin, $h_{\phi^*}$. This acts as a powerful regularization tool, reliably dampening the learning signal and preventing the policy from becoming overconfident or overfitting to saturated, less informative examples.

While we instantiate the coefficient $c(|h_{\phi^*}|)$ using the specific continuous function defined in Equation 4 for our empirical evaluation, we note that the theoretical guarantees established in Propositions 4.1 and 4.2 hold for any generalized function $c$ that satisfies the stated boundary conditions (i.e., outputting values $> 1$ for low margins and $< 1$ for high margins) and necessary regularity assumptions.

Our theoretical results establish that MADPO offers a more granular, per-pair control over the learned preference margin compared to the global mechanisms of IPO and $\beta$-DPO. Propositions 4.1 and 4.2 formalize this: for each preference pair, the optimal policy learns to match a dynamically scaled target, $c(|h_{\phi^*}|)h_{\phi^*}$, and this target margin can be monotonically controlled by the coefficient $c$. This provides a sharp, per-example comparative-statics guarantee.

In contrast, IPO regularizes globally (a uniform mechanism that does not adapt to each pair's margin), and $\beta$-DPO adapts a batch-level temperature $\beta$ shared across all samples in a batch. Neither provides the sample-specific, per-pair control formalized by Propositions 4.1 and 4.2.

## 4.2 Lipschitz Continuity and Robustness to Reward Estimation Error

Having analyzed our method in an ideal oracle setting, we now establish its robustness to the reward estimation errors that occur in practice. To do so, we prove that the MADPO loss function is Lipschitz continuous with respect to the reward model parameters, culminating in a formal performance guarantee that bounds the impact of these errors.

Our analysis in this section follows the theoretical framework established by Chowdhury et al. (2024). We adopt the following assumptions from their work to analyze the stability of our method.

**Assumption 4.3.** *We assume the following constraints on the reward model's parameter space, $\Phi$:*

- *The parameter space is defined as $\Phi = \{\phi \in \mathbb{R}^\delta \mid \sum_{i=1}^{\delta} \phi_i = 0\}$.*

- *There exists a constant $B > 0$ such that for any parameter vector $\phi \in \Phi$, its Euclidean norm is bounded: $\|\phi\| \leq B$.*

This assumption places two standard constraints on the reward model's parameter space. The first, the zero-mean condition ($\sum \phi_i = 0$), is necessary for identification. Because the preference probability in the BTL model depends only on the difference in rewards, $r_\phi(x, y_w) - r_\phi(x, y_l)$, the underlying reward function is only unique up to an arbitrary constant shift. This constraint resolves the inherent shift ambiguity, ensuring the identification of a reward function. The second condition, boundedness ($\|\phi\| \leq B$), is a standard regularity assumption required in most theoretical analyses to ensure the parameter space is well-behaved, forming a necessary prerequisite for the performance guarantees that follow.

**Assumption 4.4.** *The reward function $r_\phi(x, y)$ is assumed to be well-behaved with respect to its parameters. Specifically, there exist constants $\alpha_0, \alpha_1, \alpha_2 > 0$ such that for any $\phi \in \Phi$ and any sample $(x, y)$, the function, its gradient, and its Hessian are uniformly bounded:*

$$|r_\phi(x, y)| \leq \alpha_0, \ \|\nabla_\phi r_\phi(x, y)\| \leq \alpha_1, \ \nabla_\phi^2 r_\phi(x, y) \preceq \alpha_2 I.$$

This assumption imposes standard smoothness and boundedness conditions on the reward function, $r_\phi$. These conditions are crucial as they ensure that the explicit reward margin function, $h_\phi(x, y_w, y_l) = r_\phi(x, y_w) - r_\phi(x, y_l)$, is also well-behaved. Specifically, they imply that $h_\phi$ is bounded and Lipschitz continuous with respect to its parameters $\phi$, and that its gradient is also Lipschitz continuous. Such regularity assumptions are a common prerequisite for establishing performance guarantees in the analysis of policy optimization algorithms and are consistent with the theoretical frameworks used in related work (Agarwal et al., 2021; Chowdhury et al., 2024).

**Assumption 4.5.** *There exists a constant $L_\theta > 0$ such that for any policy parameters $\theta$ in the parameter space $\Theta$ and for any sample $(x, y_w, y_l)$, the absolute value of the log-likelihood term is bounded:*

$$|\log \sigma(\beta h_\theta(x, y_w, y_l))| \leq L_\theta.$$

This is a standard technical assumption that is well-justified. It is a mild condition, as it is fundamentally a constraint that the policy, $\pi_\theta$, cannot assign a probability of exactly 0 or 1 to any response. Furthermore, this condition is not arbitrary; it can be derived by imposing boundedness constraints on the policy function, $\pi_\theta(y|x)$, that are directly analogous to those we placed on the reward model in Assumption 4.4. This approach is consistent with similar assumptions made in prior theoretical analyses of preference-based learning, including the framework established by Chowdhury et al. (2024).

The following theorem provides a high-probability bound that connects the estimation error of the reward model to the stability of our final loss function. This error is measured in a data-dependent semi-norm, $\|\cdot\|_{\hat{\Sigma}_\phi}$, which is induced by the empirical covariance of the reward gradients.

Let the gradient vectors $\mathbf{z}_i$ be defined with respect to the reward model parameters $\phi$:

$$\mathbf{z}_i = \nabla_\phi h_\phi(x_i, y_{w,i}, y_{l,i}).$$

Then, the empirical covariance matrix $\hat{\Sigma}_\phi$ is given by:

$$\hat{\Sigma}_\phi = \frac{1}{N} \sum_{i=1}^{N} \mathbf{z}_i \mathbf{z}_i^\top.$$

**Theorem 4.6.** *Under Assumptions 4.3, 4.4 and 4.5, let $\rho \in (0,1]$ and $\kappa > 0$. Then, with a probability of at least $1 - \rho$, the following bound holds for the MADPO loss function, for all $(x, y_w, y_l) \in \mathcal{D}$*

$$\left| \mathcal{L}(\theta, \phi^*; x, y_w, y_l) - \mathcal{L}(\theta, \hat{\phi}; x, y_w, y_l) \right| \leq L \cdot \|\hat{\phi} - \phi^*\|_{\hat{\Sigma}_{\phi^*} + \kappa I}$$

$$\leq \frac{L \cdot C}{\gamma} \sqrt{\frac{\delta + \log(1/\rho)}{N}} + C' \cdot B \sqrt{\kappa + \frac{\alpha_2}{\gamma} + \alpha_1 \alpha_2 B},$$

*where $L$ is the Lipschitz constant, $C$ and $C'$ are absolute constants, and $\gamma$ is a constant dependent on the bound of the reward function:*

$$\gamma = \frac{1}{2 + e^{-4\alpha_0} + e^{4\alpha_0}}.$$

This result certifies the plug-in stability of our practical, two-stage MADPO pipeline. It guarantees that the training objective remains stable even when using an estimated reward model, $\hat{\phi}$, instead of the optimal, $\phi^*$. Concretely, Theorem 4.6 shows that for any given preference pair, the gap between the oracle loss $\mathcal{L}(\theta, \phi^*; x, y_w, y_l)$ and the plug-in loss $\mathcal{L}(\theta, \hat{\phi}; x, y_w, y_l)$ is controlled linearly by the reward-parameter error. This guarantee is per-sample and uniform over the dataset, which is a stronger claim than a statement about the average-case error. This means every mini-batch, curriculum subset, or the full empirical objective inherits the same deviation control.

The bound is also operational, as it reveals the key levers that ensure MADPO is robust in practice. The two terms in the bound expose what makes the plug-in procedure reliable:

- **Data Quality and Quantity:** The first term decays at the familiar $O(\sqrt{(\delta + \log(1/\rho))/N})$ rate with more data. It also improves with more informative data that leads to a well-conditioned covariance matrix $\hat{\Sigma}_\phi$.

- **Regularization and Boundedness:** The second term shows that gentle regularization $\kappa$ stabilizes learning in directions where data is sparse. Furthermore, the constant $\gamma$, derived from the bounded-reward assumption, prevents the logistic loss from saturating, ensuring that small errors in $\hat{\phi}$ do not cause disproportionately large swings in the objective.

It is important to note that the second term in this bound is non-vacuous in real-world settings. When the training dataset is small or the reward parameter space is sparsely sampled, this term represents the active prevention of optimization instability. By incorporating the gentle regularization $\kappa$, MADPO ensures that the learning process remains grounded even in under-sampled directions of the parameter space, making the mechanism practically relevant even for well-behaved data.

Finally, this theorem complements our earlier results. Propositions 4.1 and 4.2 characterize what MADPO learns at the pair level (amplifying low margins, shrinking high ones); Theorem 4.6 guarantees how reliably that mechanism survives the reality of an estimated reward model. In short, the pairwise control that defines MADPO is not brittle. Under the stated regularity conditions, the two-stage procedure is uniformly robust to reward estimation error, making the method dependable for the training regimes used in practice.

### 4.3 Smoothness & Curvature: MADPO vs. DPO

In this subsection, we compare the smoothness and curvature of the MADPO loss to the vanilla DPO loss. We show that the gradient and Hessian of our objective are simply a re-scaled version of the DPO derivatives, which confirms that our method has a stable and well-behaved optimization landscape.

**Proposition 4.7.** *Let $\mathcal{L}(\theta, \phi; x, y_w, y_l)$ be the MADPO loss function with bounded, $\theta$ independent weight $0 < w(h_\phi) \leq w_{max}$. Then, for any sample $(x, y_w, y_l) \in \mathcal{D}$, the first and second derivatives of the loss with respect to the implicit reward margin, $h_\theta$, satisfy the following bounds:*

*1. Bounded Gradient:*

$$\left| \frac{\partial \mathcal{L}}{\partial h_\theta} \right| \leq w_{max}\beta.$$

*2. Bounded Hessian:*

$$\left| \frac{\partial^2 \mathcal{L}}{\partial h_\theta^2} \right| \leq \frac{w_{max}\beta^2}{4}.$$

This proposition reveals that MADPO is a principled modification of the vanilla DPO framework. It shows that the scalar gradient and Hessian of the MADPO loss are simply the vanilla DPO derivatives multiplied by our bounded weight, $w(h_\phi)$. This direct scaling is crucial because it ensures MADPO preserves the benign optimization geometry of the original DPO objective, which has intrinsically capped sensitivity and curvature. For practitioners, this translates to predictable gradients that are compatible with standard control techniques like learning rate tuning or gradient clipping. Crucially, because these derivatives are bounded on a per-sample basis, it guarantees that the instance-level amplification and regularization effects established in our prior propositions are applied in a stable and reliable manner.

## 5 Experiment

In this section, we present the empirical evaluation of our proposed method, MADPO. We first outline our experimental setup on a real-world summarization task, designed to evaluate our algorithm against strong baselines using actual human preference data. We then present our main comparative results, isolate our core components through an ablation study, and empirically validate our theoretical claims regarding gradient stability. To maintain focus on the essence of these primary findings, comprehensive details regarding the experimental setup and further analysis are provided in Appendix B.

### 5.1 Experimental Setup on TL;DR summarization

**Dataset.** We use the Reddit TL;DR summarization dataset (Stiennon et al., 2020). To study the effect of data quality, we exclusively utilize pairs with explicitly provided confidence scores, discarding any instances with null values. This ensures that we accurately capture the true distribution of preference clarity, which is naturally and nearly uniformly distributed. For training, we utilize the entirety of the dataset excluding the `valid 1` split.

**Models and Evaluation.** We use `Qwen2.5-7B-Instruct` (Yang et al., 2024) as our base architecture, fine-tuning all models via LoRA (Hu et al., 2022). Following the MADPO pipeline, we first train a reward model on the training set to extract explicit margins, followed by policy optimization using these margins for adaptive weighting. We evaluate the models on 1,000 held-out pairs from the `valid 1` split using an LLM-as-a-judge framework. To rigorously mitigate position bias, evaluations are run twice with swapped summary positions; a win is only recorded upon strict agreement, and our primary metric is the net win rate (win rate minus loss rate).

**Baselines and Hyperparameters.** We compare MADPO against vanilla DPO and other variants designed to handle mixed-quality preference pairs, namely IPO and $\beta$-DPO. For all methods, the base temperature $\beta$ is tuned over the discrete range $\{0.01, 0.05, 0.1\}$. To ensure a fair comparison without the prohibitive computational cost of a full Cartesian product search on a 7B parameter model, we conduct targeted, sequential hyperparameter optimizations for both $\beta$-DPO and MADPO. All hyperparameter tuning is conducted on a separate validation subset to prevent data leakage into our final evaluation set.

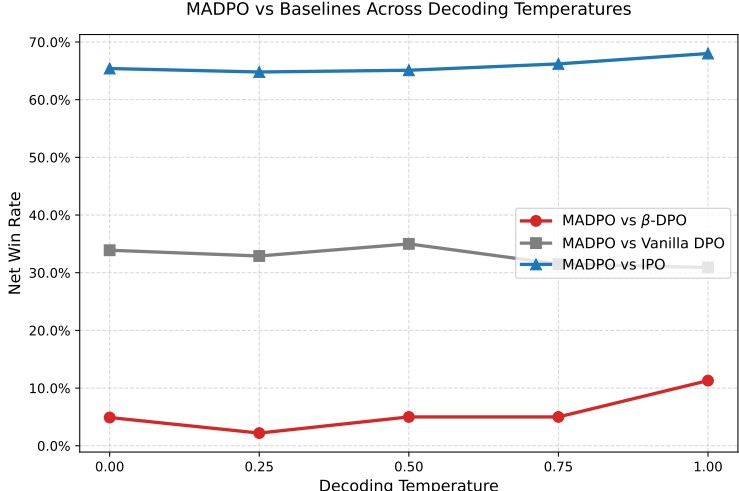

Figure 1: Net win rates of MADPO against baseline models (IPO, vanilla DPO, and $\beta$-DPO) across varying decoding temperatures. MADPO demonstrates superior robustness and consistent performance gains against all competitors.

## 5.2 Experimental Results

**Main Results.** The main experimental results, visualized in Figure 1, demonstrate that MADPO consistently outperforms all baseline methods on the TL;DR summarization task across a comprehensive range of sampling temperatures. We report the net win rate (win rate minus loss rate) as judged by the LLM oracle to establish comparative robustness.

MADPO achieves a commanding lead over IPO, maintaining net win rates between +64.8% and +68.0%. This severe underperformance by IPO suggests that its uniform regularization is overly rigid and poorly suited for the summarization task. Against vanilla DPO, MADPO maintains a highly stable and significant advantage, with net win rates holding between +30.9% and +35.0%. This confirms the core premise of our work: a single, static $\beta$ penalty is fundamentally insufficient for handling datasets with mixed-quality preference pairs.

The most competitive baseline is $\beta$-DPO, yet MADPO still secures a consistent victory across the entire temperature sweep (net win rates from +2.2% to +11.3%). Notably, at the highest sampling temperature, MADPO's advantage over $\beta$-DPO widens to its peak of +11.3%. High generation temperatures often risk inducing hallucinations or unfaithful outputs (Li et al., 2025; Spracklen et al., 2025). MADPO's strong performance in this high-entropy regime suggests that its fine-grained, instance-level regularization provides more reliable guardrails against generation degradation than the batch-level approximations used by $\beta$-DPO.

**Ablation Study.** The results of our ablation study, presented in Figure 2, confirm that the amplification mechanism is the primary driver of MADPO's massive performance gains, while the explicit regularization component remains crucial for optimizing peak deterministic performance.

To isolate the contributions of our method's two core components, we calculate the net win rates of two ablated models directly against the unaligned Qwen 2.5 7B base model (Figure 2a). The first is an Amplification-Only version, where the weight is set to one for high-margin pairs ($|h_{\hat{\phi}}| \geq \tau$) to disable the regularization component. This model achieves a massive performance leap, securing net win rates of 80.0% to 83.0% against the base model, compared to Vanilla DPO's 42.0% to 44.9%. This unequivocally demonstrates that aggressively learning from informative, low-margin pairs is the most critical factor for success in real world, noisy datasets.

The second model is a Regularization-Only version, where the weight is set to one for low-margin pairs ($|h_{\hat{\phi}}| < \tau$) to disable amplification. While its impact is secondary, explicitly dampening high-margin pairs still

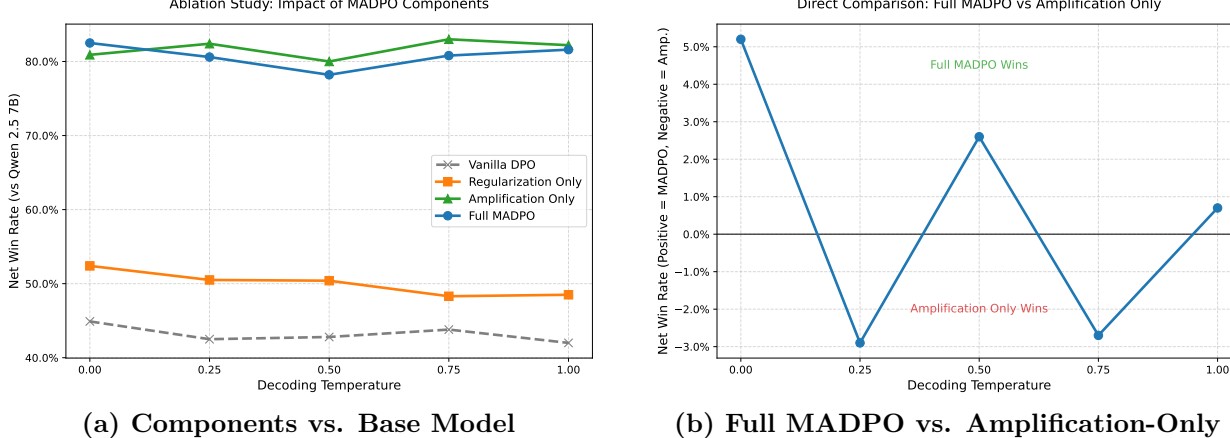

**(a) Components vs. Base Model**   **(b) Full MADPO vs. Amplification-Only**

Figure 2: Ablation study evaluating the impact of MADPO's core components. **(a)** Net win rates of the ablated models and Vanilla DPO evaluated against the unaligned Qwen 2.5 7B base model across decoding temperatures. **(b)** Head-to-head net win rate of Full MADPO evaluated directly against the Amplification-Only model, highlighting Full MADPO's significant advantage at the deterministic $T = 0$ setting.

provides a consistent improvement (net win rates of 48.3% to 52.4%) over the Vanilla DPO baseline across all temperatures, confirming that the regularization component is a beneficial mechanism in its own right.

Crucially, Figure 2b addresses whether the explicit regularization component is still necessary given the dominance of the amplification mechanism. In a direct head-to-head comparison across higher sampling temperatures ($\geq 0.25$), Full MADPO and the Amplification-Only model effectively trade marginal fluctuations, representing a statistical tie. However, with greedy decoding, which is the standard and most critical setting for deploying reliable summarization models (Li et al., 2025), Full MADPO secures a significant +5.2% net win rate over the Amplification-Only model. This confirms that the Full MADPO framework is highly meaningful: the explicit regularization component acts as a necessary safeguard against overfitting on easy examples, ensuring optimal deterministic generation without sacrificing the substantial gains provided by amplification.

**Empirical Gradient Stability.** Finally, we design a stress test to empirically validate the theoretical guarantees of Proposition 4.7, which asserts that MADPO preserves a stable, bounded optimization landscape. To simulate a worst case adversarial scenario, we synthesize a corrupted dataset from the Reddit TL;DR corpus. We isolate 30% of the training pairs that exhibit a large estimated reward margin ($h_\phi > \tau$) and deliberately invert their preference labels. This forces these pairs into the extreme negative margin regime ($h_\phi < -\tau$). The remaining 70% of the data is left in its normal, correct state.

It is important to note the objective of this experimental design. The 30% label inversion is not intended to model realistic human annotation noise; rather, it is strictly an extreme worst-case scenario designed to empirically verify the mathematical stability guarantees derived in Proposition 4.7. By intentionally pushing the optimization landscape to an adversarial extreme, we isolate and prove MADPO's structural resilience to gradient explosion.

Using the fixed parameters at $c_{\max} = 4$ and $\tau = 7$, we track the gradient $L_2$ norm during training. Crucially, we disable all gradient clipping mechanisms to observe the raw optimization dynamics. We compare three setups: Vanilla DPO, our proposed MADPO (which applies the piecewise cap defined in Equation 4), and an Uncapped MADPO variant where the protective piecewise boundary is removed.

The results, shown in Figure 3, starkly illustrate the necessity and effectiveness of our design. Without the piecewise constraint, Uncapped MADPO suffers from highly unstable optimization; its gradient norm spikes by orders of magnitude as it exponentially amplifies the adversarial, negative-margin errors. In stark contrast, the Full MADPO algorithm completely mitigates this instability. By seamlessly engaging the cap

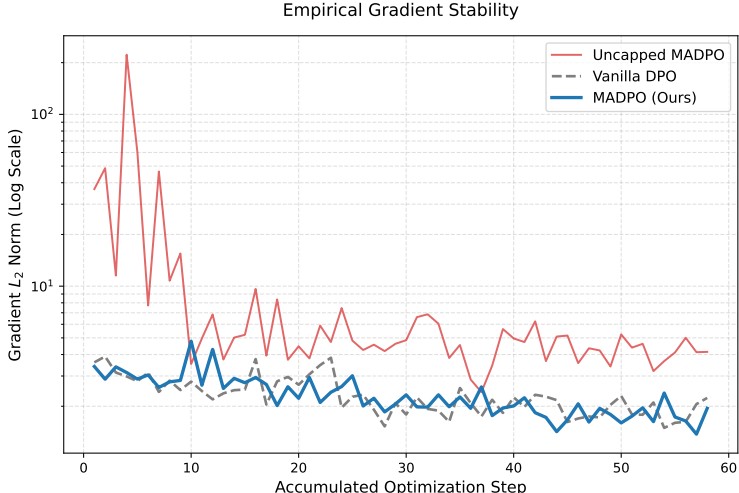

Figure 3: Empirical stress test of gradient stability on the Qwen 2.5 7B model without gradient clipping. The dataset contains 30% adversarial (inverted) labels. Uncapped MADPO exhibits severe optimization instability, whereas our proposed MADPO (using the piecewise cap from Equation 4) maintains bounded, stable gradients perfectly aligned with Vanilla DPO, validating Proposition 4.7.

from Equation 4, MADPO bounds these errant gradients, resulting in a stable optimization trajectory that closely mirrors Vanilla DPO. This confirms that MADPO achieves its aggressive performance gains without sacrificing training reliability, even when confronted with severe data noise.

## 6   Conclusion

In this work, we addressed a key limitation in the vanilla Direct Preference Optimization (DPO) framework: its reliance on a single, fixed temperature parameter that struggles with preference data of varying quality. We introduced Margin-Adaptive Direct Preference Optimization (MADPO), a method that applies an instance-level, adaptive weight to the DPO loss. Our theoretical analysis proved that MADPO is a principled modification that maintains a stable optimization landscape and is robust to the errors inherent in practical two-step training pipelines. Our empirical results on a TL;DR summarization task confirmed these findings, demonstrating that MADPO consistently outperforms strong baselines across various sampling temperatures. Our analysis reveals this success stems from effectively amplifying informative, low-margin examples while safely regularizing saturated pairs.

We acknowledge two primary limitations in the current study. First, although we scaled our experiments to a 7B-parameter model using LoRA, all results are still obtained on a single model family and a single summarization task. We therefore cannot claim that the observed gains generalize to other large language models or to substantially different domains. Second, while we used data with real human-annotated preference data, the method has only been evaluated on one type of preference task. The behavior of MADPO on more diverse or noisier real-world preference distributions remains to be thoroughly investigated.

Importantly, we do not claim that MADPO is universally superior to existing methods; our goal is to demonstrate a principled and stable approach to instance-level margin adaptation within the DPO framework. Broader validation across model scales, architectures, and tasks is left for future work.

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

## A   Proofs

*Proof of Proposition 4.1.* For mathematical representation in this proof, we adopt the notation $(x, y, y', d) \in \mathcal{D}$, where $y, y'$ are responses and $d \in \{0, 1\}$ is a binary preference indicator such that $d = 1$ indicates $y \succ y'$ (i.e., $y = y_w$, $y' = y_l$) and $d = 0$ indicates $y' \succ y$ (i.e., $y' = y_w$, $y = y_l$). The introduction of $d$ allows us to flexibly model preference directions using the BTL model, where $E_{d \sim P_{\phi^*}}[d|x, y, y'] = \sigma(h_{\phi^*})$. We now analyze the expected loss for any triplet $(x, y, y', d) \in \mathcal{D}_{\text{low}}$. The analysis begins by taking the expectation of the sample-level loss over the optimal preference distribution $d \sim P_{\phi^*}$. This expectation simplifies to a new cross-entropy objective.

The derivation proceeds as follows:

$$
\begin{aligned}
\mathcal{L}(\pi_\theta, \phi^*; x, y, y') &= -\mathbb{E}_{d \sim P_{\phi^*}} \left[ d \cdot w(h_{\phi^*}) \log \sigma(\beta h_\theta) + (1 - d) \cdot w(-h_{\phi^*}) \log \sigma(-\beta h_\theta) | x, y, y' \right] \\
&= - \left[ \sigma(h_{\phi^*}) w(h_{\phi^*}) \log \sigma(\beta h_\theta) + \sigma(-h_{\phi^*}) w(-h_{\phi^*}) \log \sigma(-\beta h_\theta) \right] \\
&= - \left[ \sigma(c(|h_{\phi^*}|) \cdot h_{\phi^*}) \log \sigma(\beta h_\theta) + \sigma(-c(|h_{\phi^*}|) \cdot h_{\phi^*}) \log \sigma(-\beta h_\theta) \right].
\end{aligned}
$$

The second equality holds by BTL where $E_{d \sim P_{\phi^*}}[d|x, y, y'] = \sigma(h_{\phi^*})$. The last equality is satisfied by the construction of $w(h)$. This final form is the cross-entropy loss between the policy's distribution, $P_\theta$, and a margin-aware target distribution, $P'_{\phi^*}$, with logits defined by $c(|h_{\phi^*}|) \cdot h_{\phi^*}$. Since the total loss, $\mathcal{L}(\theta, \phi^*)$, is the expectation of these non-negative, instance-level cross-entropy losses over the dataset, the global minimum is achieved when the loss for each instance is minimized simultaneously. This occurs if and only if the distributions are identical for each instance, which requires their logits to be equal. Therefore, the optimal solution satisfies:

$$
\beta h_{\theta^*} = c(|h_{\phi^*}|) \cdot h_{\phi^*}.
$$

$\square$

*Proof of Proposition 4.2.* As we did for the proof of Proposition 4.1, we adopt the notation $(x, y, y', d) \in \mathcal{D}$. Without loss of generality, we assume $h_{\phi^*} \equiv h_{\phi^*}(x, y, y') = r_{\phi^*}(x, y) - r_{\phi^*}(x, y') > 0$ by swapping $y$ and $y'$ if necessary. So the weighting function is

$$
w(h_{\phi^*}) = \frac{\sigma(c(h_{\phi^*}) \cdot h_{\phi^*})}{\sigma(h_{\phi^*})} \quad \text{and} \quad w(-h_{\phi^*}) = 1
$$

We derive the expected loss for a triplet $(x, y, y', d) \in \mathcal{D}_{\text{high}}$ with $E_{d \sim P_{\phi^*}}[d|x, y, y'] = \sigma(h_{\phi^*})$ from BTL model:

$$
\mathcal{L}(\pi_\theta, \phi^*; x, y, y') = - \left[ \sigma(c(|h_{\phi^*}|) \cdot h_{\phi^*}) \log \sigma(\beta h_\theta) + \sigma(-h_{\phi^*}) \log \sigma(-\beta h_\theta) \right]. \tag{6}
$$

Now, we take derivatives of Eq. equation 6 with respect to $\beta h_\theta$, which results in

$$
\frac{\partial \mathcal{L}(\pi_\theta, \phi^*; x, y, y')}{\partial \beta h_\theta} = - \left[ \sigma(c(|h_{\phi^*}|) \cdot h_{\phi^*}) \sigma(-\beta h_\theta) - \sigma(-h_{\phi^*}) \sigma(\beta h_\theta) \right].
$$

At the optimum $\theta^*$, we have

$$
F(c, \beta h_{\theta^*}) \equiv \left. \frac{\partial \mathcal{L}(\pi_\theta, \phi^*; x, y, y')}{\partial \beta h_\theta} \right|_{\theta = \theta^*} = - \left[ \sigma(c(|h_{\phi^*}|) \cdot h_{\phi^*}) \sigma(-\beta h_{\theta^*}) - \sigma(-h_{\phi^*}) \sigma(\beta h_{\theta^*}) \right] = 0.
$$

We denote $c \equiv c(|h_{\phi^*}|)$ for notation simplicity. By the implicit function theorem, $\frac{\partial \beta h_{\theta^*}}{\partial c} = -\frac{\partial F/\partial c}{\partial F/\partial \beta h_{\theta^*}}$. Compute the partial derivatives:

$$
\begin{aligned}
\frac{\partial F}{\partial c} &= \frac{\partial}{\partial c} \left[ \sigma(c \cdot h_{\phi^*}) \sigma(-\beta h_{\theta^*}) - \sigma(-h_{\phi^*}) \sigma(\beta h_{\theta^*}) \right] \\
&= h_{\phi^*} \sigma(c \cdot h_{\phi^*}) \sigma(-c \cdot h_{\phi^*}) \sigma(-\beta h_{\theta^*}). \\
\frac{\partial F}{\partial \beta h_{\theta^*}} &= \sigma(c \cdot h_{\phi^*}) \cdot \left[ -\sigma(\beta h_{\theta^*}) \sigma(-\beta h_{\theta^*}) \right] - \sigma(-h_{\phi^*}) \cdot \left[ \sigma(\beta h_{\theta^*}) \sigma(-\beta h_{\theta^*}) \right] \\
&= -\sigma(\beta h_{\theta^*}) \sigma(-\beta h_{\theta^*}) \left[ \sigma(c \cdot h_{\phi^*}) + \sigma(-h_{\phi^*}) \right].
\end{aligned}
$$

Thus:

$$\frac{\partial \beta h_{\theta^*}}{\partial c} = -\frac{h_{\phi^*}\sigma(c \cdot h_{\phi^*})\sigma(-c \cdot h_{\phi^*})\sigma(-\beta h_{\theta^*})}{-\sigma(\beta h_{\theta^*})\sigma(-\beta h_{\theta^*})\left[\sigma(c \cdot h_{\phi^*}) + \sigma(-h_{\phi^*})\right]} = \frac{h_{\phi^*}\sigma(c \cdot h_{\phi^*})\sigma(-c \cdot h_{\phi^*})\sigma(-\beta h_{\theta^*})}{\sigma(\beta h_{\theta^*})\sigma(-\beta h_{\theta^*})\left[\sigma(c \cdot h_{\phi^*}) + \sigma(-h_{\phi^*})\right]}.$$

Since $h_{\phi^*} > 0$, and all sigmoid terms are strictly positive, the numerator and denominator are positive, so $\frac{\partial \beta h_{\theta^*}}{\partial c} > 0$. Thus, $\frac{\partial(\beta h_{\theta^*})}{\partial c} > 0$, proving the proposition. $\qquad\square$

**Lemma 1.** *Let $w(h)$ be the weight function defined in Eq. 4. The function is differentiable almost everywhere, and its derivative, $w'(h)$, is uniformly bounded. That is, there exists a constant $L_w > 0$ such that for all $h \in \mathbb{R}$ where the derivative is defined:*

$$|w'(h)| \leq L_w.$$

*Proof.* For the purpose of this analysis, we can simplify the expression by fixing the hyperparameters to representative values. Hyperparameters' specific value does not affect the following stability analysis, so for simplicity we let $c_{\min} = 0$, along with $c_{\max} = 2$ and $\lambda = 1$.

By construction, the weight function $w(h)$ is continuous everywhere and differentiable almost everywhere. The only non differentiable points occur at $h = 0$ due to the absolute value in the coefficient function $c$ and at $h = -\tau$ due to the piecewise definition.

- **Case 1: For $h \in (0, \infty)$.**
  Let $k(h) = c(h) \cdot h$. The weight function and its derivative are:

  $$w(h) = \frac{\sigma(k(h))}{\sigma(h)}, \quad w'(h) = \frac{\sigma'(k(h))k'(h)\sigma(h) - \sigma(k(h))\sigma'(h)}{\sigma^2(h)}$$

  The sigmoid function $\sigma(\cdot)$ and its derivative $\sigma'(\cdot)$ are universally bounded (by 1 and 1/4, respectively). For $h > 0$, the denominator $\sigma^2(h)$ is bounded away from zero. Therefore, to show that $w'(h)$ is bounded, we only need to show that $k'(h)$ is bounded.

  By the product rule, $k'(h) = c(h) + c'(h)h$. By definition, $c(h)$ is bounded between $[c_{\min}, c_{\max}]$. The term $c'(h)h$ involves the derivative of the coefficient function, which contains an exponential decay term that dominates the linear growth of $h$. As $h \to \infty$, $c'(h)h \to 0$. Since $k'(h)$ is continuous on $(0, \infty)$ and has finite limits at its boundaries ($h \to 0^+$ and $h \to \infty$), it is bounded on this interval. Thus, $w'(h)$ is also bounded for $h > 0$.

- **Case 2: For $h \in (-\tau, 0)$.**
  On this bounded open interval, the derivative $w'(h)$ is a continuous function. As established in our analysis of the boundaries, the one-sided limits of $w'(h)$ as $h \to -\tau^+$ and as $h \to 0^-$ are both finite. A function that is continuous on a bounded open interval and has finite limits at its endpoints is necessarily bounded. Thus, $w'(h)$ is bounded on this interval.

- **Case 3: For $h < -\tau$.**
  In this region, $w(h) = 1$. Therefore, its derivative is $w'(h) = 0$, which is trivially bounded.

Since the derivative $w'(h)$ is bounded on all three regions that cover its domain, we conclude that it is uniformly bounded. $\qquad\square$

**Lemma 2.** *The weight function $w$ defined in Eq. 4 is $L_w$-Lipschitz continuous. That is, for any $h, h' \in \mathbb{R}$, the following inequality holds:*

$$|w(h) - w(h')| \leq L_w|h - h'|. \tag{7}$$

*Proof.* The proof relies on Lemma 1, which establishes that the derivative of the weight function is bounded almost everywhere, i.e., $|w'(t)| \leq L_w$. A function with a bounded derivative (a.e.) is absolutely continuous, which is the required condition to apply the Fundamental Theorem of Calculus for Lebesgue Integrals.

For any two points $a, b \in \mathbb{R}$ with $a < b$, the theorem states:

$$w(b) - w(a) = \int_a^b w'(t)dt. \tag{8}$$

We can now take the absolute value of both sides and apply the bound from our lemma:

$$
\begin{aligned}
|w(b) - w(a)| = \left| \int_a^b w'(t)dt \right| \\
\leq \int_a^b |w'(t)|dt \qquad &\text{(Triangle inequality for integrals)} \\
\leq \int_a^b L_w dt \qquad &\text{(By Lemma 1, } |w'(t)| \leq L_w) \\
= L_w(b - a). &
\end{aligned}
$$

Since this holds for any $a < b$, we can generalize to $|w(h) - w(h')| \leq L_w |h - h'|$ for any $h, h' \in \mathbb{R}$. This is the definition of $L_w$-Lipschitz continuity. $\qquad \square$

*Proof of Theorem 4.6.* The proof proceeds in two main parts. First, we establish that the margin function $h_\phi$ is Lipschitz continuous with respect to its parameters $\phi$. Second, we leverage this result to show that the full loss function, $\mathcal{L}(\theta, \phi; x, y_w, y_l)$, is also Lipschitz continuous, which allows us to invoke the final result from prior work.

**Part 1: Lipschitz Continuity of the Margin Function ($h_\phi$).** We begin with the definition of the margin function: $h_\phi(x, y_w, y_l) = r_\phi(x, y_w) - r_\phi(x, y_l)$. By the Mean Value Theorem, there exists a $\bar{\phi}$ on the line segment between $\phi^*$ and $\hat{\phi}$ such that:

$$h_{\hat{\phi}}(x, y_w, y_l) - h_{\phi^*}(x, y_w, y_l) = \left( \nabla_\phi r_{\bar{\phi}}(x, y_w) - \nabla_\phi r_{\bar{\phi}}(x, y_l) \right)^\top (\hat{\phi} - \phi^*).$$

Taking the absolute value and applying the generalized Cauchy-Schwarz inequality with the semi-norm $\| \cdot \|_{\Sigma_\phi + \kappa I}$ gives:

$$|h_{\hat{\phi}} - h_{\phi^*}| \leq \left\| \nabla_\phi r_{\bar{\phi}}(x, y_w) - \nabla_\phi r_{\bar{\phi}}(x, y_l) \right\|_{(\Sigma_\phi + \kappa I)^{-1}} \cdot \|\hat{\phi} - \phi^*\|_{\Sigma_\phi + \kappa I}.$$

Under Assumption 4.4, the gradient of the reward function is bounded. This implies that the term involving the gradients is also bounded by some constant, which we will call $L_\phi$. Thus, the margin function is $L_\phi$-Lipschitz continuous with respect to $\phi$:

$$|h_{\hat{\phi}}(x, y_w, y_l) - h_{\phi^*}(x, y_w, y_l)| \leq L_\phi \|\hat{\phi} - \phi^*\|_{\Sigma_\phi + \kappa I}.$$

**Part 2: Lipschitz Continuity of the Full Loss Function ($\mathcal{L}$).** Now we analyze the full loss function, $\mathcal{L}(\theta, \phi; x, y_w, y_l) = -w(h_\phi(x, y_w, y_l)) \log \sigma(\beta h_\theta(x, y_w, y_l))$.

$$
\begin{aligned}
\left| \mathcal{L}(\theta, \phi^*; x, y_w, y_l) - \mathcal{L}(\theta, \hat{\phi}; x, y_w, y_l) \right| &= |-\log \sigma(\beta h_\theta(x, y_w, y_l)| \cdot \left| w(h_{\phi^*}(x, y_w, y_l)) - w(h_{\hat{\phi}}(x, y_w, y_l)) \right| \\
&\leq L_\theta \cdot \left| w(h_{\phi^*}(x, y_w, y_l)) - w(h_{\hat{\phi}}(x, y_w, y_l)) \right| \\
&\leq L_\theta \cdot L_w \cdot |h_{\phi^*}(x, y_w, y_l) - h_{\hat{\phi}}(x, y_w, y_l)| \\
&\leq L_\theta \cdot L_w \cdot L_\phi \cdot \|\hat{\phi} - \phi^*\|_{\Sigma_\phi + \kappa I}.
\end{aligned}
$$

The second inequality holds by Assumption 4.5. The third inequality holds by Lemma 2. The last inequality holds by Part 1 of this proof. By defining the final Lipschitz constant as $L = L_\theta L_w L_\phi$, we have shown that our loss function is L-Lipschitz continuous:

$$\left| \mathcal{L}(\theta, \phi^*; x, y_w, y_l) - \mathcal{L}(\theta, \hat{\phi}; x, y_w, y_l) \right| \leq L \cdot \|\hat{\phi} - \phi^*\|_{\Sigma_\phi + \kappa I}.$$

With this result established, the final statistical bound on the estimation error follows directly by invoking the general framework for two-step estimators, such as Theorem 4.2 in Chowdhury et al. (2024). This completes the proof. □

*Proof of Proposition 4.7.* For readability in this proof, we suppress the explicit dependence on the sample $(x, y_w, y_l)$ and parameters $(\theta, \phi)$ for functions like $\mathcal{L}$, $h_\theta$, and $h_\phi$, unless required for clarity. The proof for both claims relies on the fact that the weight function, $w(h_\phi)$, is uniformly bounded. From its construction in Eq. 4, there exists a constant $w_{\max}$ such that $|w(h_\phi)| \leq w_{\max}$ for all inputs.

1. **Bounded Gradient:** The first derivative of the loss is given by:

$$\frac{\partial \mathcal{L}}{\partial h_\theta} = -w(h_\phi) \cdot \beta\sigma(-\beta h_\theta).$$

   This expression is a product of the weight function, a constant $\beta$, and the sigmoid function, all of which are bounded. Therefore, their product is uniformly bounded.

2. **Bounded Hessian:** The second derivative of the loss is given by:

$$\frac{\partial^2 \mathcal{L}}{\partial h_\theta^2} = w(h_\phi) \cdot \beta^2\sigma(-\beta h_\theta)\sigma(\beta h_\theta).$$

   This is a product of the bounded weight function, a constant $\beta^2$, and the derivative of the sigmoid function, $\sigma'(\cdot) = \sigma(\cdot)\sigma(-\cdot)$, which is famously bounded by $1/4$. Therefore, the second derivative is also uniformly bounded.

□

# B   Detailed Experiment

## B.1   Detailed Experimental Setup

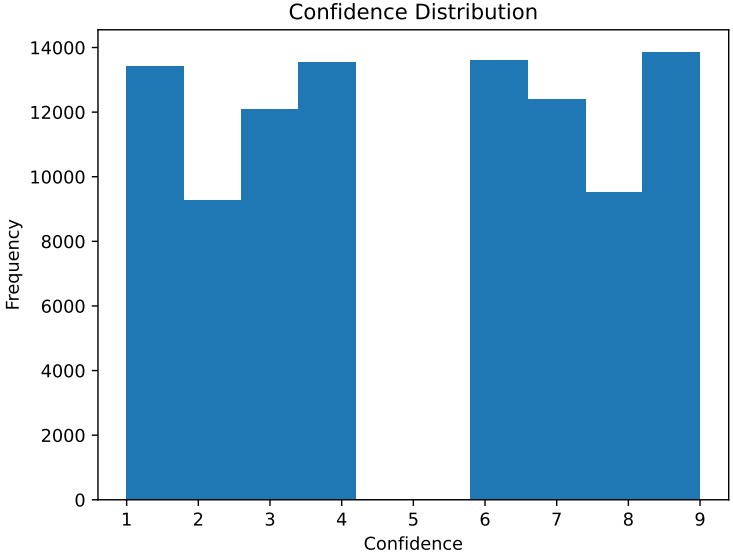

Figure 4: Distribution of confidence scores for the filtered Reddit TL;DR training dataset. The distribution is roughly uniform across non neutral values (1–4 and 6–9), naturally exposing the models to a diverse, realistic mixture of both high agreement and low-agreement preference pairs.

**Dataset Filtering and Splits.** For our experiments using the Reddit TL;DR summarization dataset (Stiennon et al., 2020), we exclude any preference pairs lacking a recorded confidence score. This filtering step is crucial to ensure our models train on a verifiable, realistic distribution of preference clarity. As illustrated in Figure 4, the resulting data demonstrates a roughly uniform distribution across all non neutral confidence levels (1–4 and 6–9). After filtering, our final training corpus comprises 97,712 pairs, incorporating all standard data splits with the explicit exception of `valid 1`. To form our fixed evaluation set, we sample first 1,000 held-out preference pairs exclusively from this `valid 1` split.

**Model Implementation.** All stages of our experiment leverage the `Qwen2.5-7B-Instruct` (Yang et al., 2024) model. Following the MADPO two-stage pipeline, we first train a standard reward model using LoRA on the 97,712-pair training set to estimate explicit reward margins ($h_\phi$). Subsequently, we fine-tune a separate policy model using these static margins to compute the adaptive instance-level weights.

**LLM-as-a-Judge Evaluation Protocol.** We evaluate the aligned policies using Gemini 3 Flash as an impartial judge. To assess generative robustness, the policy responses are generated across a sweep of sampling temperatures $T \in \{0.0, 0.25, 0.5, 0.75, 1.0\}$. The Gemini judge evaluates these responses using a decoding temperature of 0 and a disabled thinking budget to ensure deterministic, reproducible verdicts. The judge is instructed to evaluate the summaries based on accuracy and conciseness using the following prompt:

> Which of the following is a better TL;DR summary of the Reddit post?
>
> Evaluate the summaries based on how accurately and concisely they capture the essence of the original post.
>
> Do not favor a summary because it appears first or second.
>
> Post: {post}
> Summary A: {summary_m1}
> Summary B: {summary_m2}
>
> FIRST, provide a one-sentence comparison of the two summaries.
> SECOND, state "A", "B", or "Tie" to indicate your choice.

To eliminate position bias, every comparison is executed twice, swapping the positions of Summary A and Summary B. The policy is credited with a win only if the judge consistently selects its generation across both permutations. Any discrepancy between the two evaluations defaults to a "Tie". Final performance is reported as the net win rate.

Table 1: Hyperparameters for Qwen 2.5 7B Policy Optimization.

| Hyperparameter | Value |
|---|---|
| Optimizer | AdamW |
| Learning Rate | $2 \times 10^{-5}$ |
| Adam Betas ($\beta_1, \beta_2$) | (0.9, 0.95) |
| Optimizer Epsilon | $1 \times 10^{-5}$ |
| Weight Decay | 0.01 |
| Learning Rate Schedule | Cosine Decay |
| Warm-up Ratio | 0.03 (3% of total steps) |
| Total Epochs | 1 |
| Per-Device Batch Size | 2 |
| Gradient Accumulation Steps | 16 |
| LoRA Rank ($r$) | 32 |
| LoRA Alpha ($\alpha$) | 64 |
| LoRA Dropout | 0.05 |
| LoRA Target Modules | q, k, v, o, gate, up, down (All linear layers) |

**Training Hyperparameters** To ensure reproducibility, we provide the full set of hyperparameters used for the Qwen 2.5 7B policy optimization in Table 1. All models (Reward Model, Vanilla DPO, $\beta$-DPO, and MADPO) were trained using the same base configuration to ensure a fair comparison.

## B.2 Omitted Analysis

**Sensitivity Analysis.** To complement our main results, we conduct a sensitivity analysis on MADPO's two key hyperparameters: the margin threshold ($\tau$) and the amplification intensity ($c_{\max}$). To ensure a strictly diagnostic evaluation that isolates the mathematical impact of these variables, we fix the base temperature at $\beta = 0.1$ for all models for this analysis.

Furthermore, to ensure our results are not obscured by sampling noise, we restrict this sensitivity analysis entirely to greedy decoding. Evaluating at higher stochastic temperatures introduces sampling variance that can obfuscate the pure alignment dynamics we aim to measure. By removing this variance and controlling the base temperature, we can accurately observe how stable and predictable our novel components are when evaluated against the unaligned Qwen 2.5 7B base model.

**Sensitivity to Margin Threshold ($\tau$).** We first evaluate the sensitivity of the margin threshold by varying $\tau \in \{3, 5, 7, 10\}$ while keeping the amplification intensity fixed at $c_{max} = 4$. Figure 5 (Left) demonstrates that while the trend is non-monotonic, MADPO's performance remains robustly high across the entire sweep, never dropping below a 40% net win rate against the base model. The fluctuations suggest a delicate balance in the real-world dataset: setting $\tau$ requires identifying enough "hard" pairs to amplify without over-classifying inherent noise as informative signal. Nonetheless, the consistently high absolute performance proves that MADPO is not brittle to its threshold configuration.

**Sensitivity to Amplification Intensity ($c_{max}$).** Next, we examine the impact of the amplification intensity by varying $c_{max} \in \{2, 4, 6\}$ while keeping the margin threshold fixed at $\tau = 7$. As illustrated in Figure 5 (Right), we observe a strict, monotonic increase in performance as the intensity grows. The net win rate scales from approximately 39.0% at $c_{max} = 2$ up to 58.7% at $c_{max} = 6$. This predictable, linear trend reinforces the primary conclusion of our ablation study: aggressively amplifying the learning signal for hard, low-margin preference pairs is a highly stable and effective driver of alignment performance.

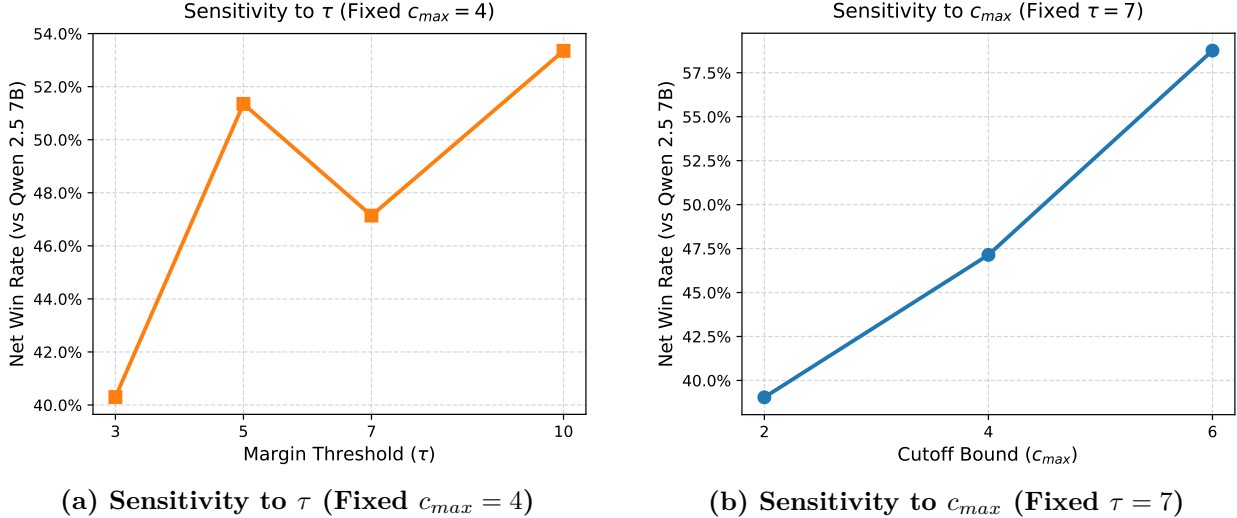

**(a) Sensitivity to $\tau$ (Fixed $c_{max} = 4$)**        **(b) Sensitivity to $c_{max}$ (Fixed $\tau = 7$)**

Figure 5: Diagnostic sensitivity analysis of MADPO's key hyperparameters evaluated with greedy decoding against the unaligned Qwen 2.5 7B base model. **(a)** Performance remains robustly high across various margin thresholds ($\tau$). **(b)** Performance scales monotonically with increased amplification intensity ($c_{max}$).

## C  Related Work

Recent research in preference alignment has sought to address the limitations of DPO's fixed temperature, which can lead to overfitting on easy, high-margin pairs. Prominent approaches such as IPO (Azar et al., 2024) and $\beta$-DPO (Wu et al., 2024b) tackle this problem with mechanisms that are applied at a coarse granularity. IPO proposes a uniform target margin for all samples, while $\beta$-DPO's strategies operate at the batch level.

While an improvement, the batch-level mechanisms of $\beta$-DPO have several notable drawbacks. First, its adaptation is a coarse approximation of the instance-level ideal. A single training batch can easily contain a mix of high- and low-margin pairs, yet the $\beta$-batch method applies a single, compromised temperature to all of them. Second, the linear adaptation rule, $\beta_{\text{batch}} = \beta_0(1 + m \cdot (\bar{h}_\theta - h_0))$, can be unstable; for difficult batches where the average margin $\bar{h}_\theta$ is negative, the resulting temperature $\beta_{\text{batch}}$ can also become negative, leading to a divergent objective that actively learns to prefer the dispreferred response. Finally, the batch-dependent temperature complicates hyperparameter selection, as the absence of a fixed $\beta$ makes it difficult to perform reliable cross-validation to find the optimal tuning parameters.

Our work, MADPO, is distinct in that it provides a fully instance-level and data-preserving solution that avoids these issues. By applying a continuous, adaptive weight to each training sample based on its unique reward margin, our method can granularly control the learning signal. This allows it to be aggressive on hard pairs and conservative on easy ones within the same batch, providing a more flexible and stable approach to preference alignment.

Beyond the methods discussed above, other recent works have extended the DPO framework in various directions. For instance, methods like SimPO (Meng et al., 2024) and $\alpha$-DPO (Wu et al., 2024a) focus on simplifying the objective by removing the need for an explicit reference policy. Another line of work has explored reweighting preference data. Omni-DPO (Peng et al., 2025) dynamically weights pairs based on both their inherent quality and the model's current learning state, while WPO (Zhou et al., 2024) reweights off-policy preference data to more closely resemble the on-policy distribution. While these methods also use a reweighting scheme, their motivation is distinct from that of MADPO. Whereas these approaches weight data based on the policy's dynamic state or distributional properties, MADPO's weighting is based on a static, external signal of sample difficulty derived from the reward margin, $h_\phi$. Our goal is not to correct for distributional shift, but to granularly control the regularization strength for each individual sample based on its intrinsic difficulty.

Finally, our work is situated within the offline preference optimization paradigm, which is distinct from online reinforcement learning methods such as PPO (Ouyang et al., 2022) and GRPO (Shao et al., 2024). While online methods can achieve high performance by sampling directly from the policy to receive real-time rewards, they entail significant computational overhead and are often prone to training instability. MADPO is motivated by the need for a more practical, sampling-free approach that effectively handles mixed-quality human preference datasets without the need for expensive environment interaction or complex actor-critic architectures. By providing a stable, instance-level solution for static data, MADPO offers a resource-efficient alternative that prioritizes data-centric robustness over the iterative sampling required by online RL frameworks.

