# OpenReview forum: "Margin Adaptive DPO: Leveraging Reward Model for Granular Control in Preference Optimization"
_TMLR — Rejected by TMLR_

### Review · Reviewer_PXMR · 2025-10-31

**Summary Of Contributions:**

The paper propose a solution to the issue of DPO introduced by the temperature, through two steps. First, a standard reward model is trained on the preference dataset to produce an explicit, per-sample estimate of the preference margin. Then this margin is used to compute an instance-level weight to adaptively modulate the standard DPO loss function. This weighting scheme is designed to achieve two complementary goals: for preference pairs with a small, subtle margin the weight amplifies the loss, forcing the model to learn more aggressively; conversely, for pairs with a large, obvious margin the weight dampens the loss, providing a regularizing effect that prevents overconfidence and overfitting.

**Audience:**

Yes

**Audience Explanation:**

Researchers in LLM and alignment will be interested in this work.

**Claims And Evidence:**

No

**Claims Explanation:**

1. The paper claims the MADPO to be a universal method, but the experiment verification is limited to a small model, and the large models are not considered (like Llama 3, deepseek).
2. The paper assumes a sentiment classifier to provide reward model for input text. However, this oracle seems impractical to me, as the human preference is more complicated. Is this assumption feasible?
3. The paper claims it obtains a more robust approach. In my view, the major robustness should be its performance on large-scale models trained on noisy preference data, where the "ground-truth" reward margin is an unknown and potentially ill-defined quantity. The paper makes a compelling case for the principle, but the evidence for its general practice is yet to be established. MADPO heavily relies on the assumption that the estimated reward model, $r_{\hat{\phi}}$, provides a faithful and well-calibrated measure of the true preference margins. What if the underlying preference cannot be modeled by a reward?
4. The advantage of DPO is it bypasses reward estimation, which seems to be needed in MADPO.

**Requested Changes:**

While the paper presents a novel and theoretically sound method in MADPO, its claims of general superiority and robustness are not fully supported by the current empirical evidence. The following revisions are requested to strengthen the manuscript by providing a more balanced and transparent assessment of the method's contributions and limitations.

1. Re-evaluating the claims of generalizability and robustness: (1). Specify the definition of robustness and verify the robustness; (2). Evaluate on larger models.
2. Experiment setup: it will be good to have additional experiments with noisy reward or even general preference.
3. Clarify the questions I mentioned above.

---

> ### Author Response · Authors · 2026-04-20
> **Author Response and Summary of Revisions**
>
> We thank the reviewer for the constructive feedback. We have significantly updated the manuscript to address concerns regarding scalability and the realism of the experimental setup. Most notably, we have replaced the initial sentiment generation task with a modern benchmark: Qwen 2.5 (7B) fine-tuned using LoRA on the Reddit TL;DR Summarization dataset. We have also clarified the theoretical scope of our robustness claims and the standard assumptions used in our framework.
>
> Below is our response to specific points:
>
> * Scalability with LoRA: To address generalizability, we moved beyond the 270M model. We now validate MADPO using Qwen 2.5 (7B) fine-tuned with LoRA on the Reddit TL;DR Summarization benchmark. As shown in Figure 1, MADPO consistently outperforms DPO, IPO, and $\beta$-DPO at this scale.
>
> * Definition of Robustness: We claim robustness specifically regarding reward model estimation error. This is proven theoretically in Theorem 4.6, which shows the MADPO loss is Lipschitz continuous with respect to reward parameters, bounding the impact of errors when the true margin is estimated. We have dropped the previous claim that MADPO is "more robust than $\beta$-DPO" to avoid confusion.
>
> * Bradley-Terry Assumption: Our work operates under the standard Bradley-Terry-Luce (BTL) framework. This is the mathematical foundation for most modern alignment literature, including RLHF and DPO. While cases where this assumption fails are important, they are outside the scope of this paper.
>
> * Offline Efficiency: While MADPO requires a reward modeling step, it preserves the main advantage of DPO by bypassing Online RL (PPO) and its associated sampling costs. Since training an offline Reward Model is significantly cheaper than training the Policy Model, the performance gains over DPO justify this step.

---

### Review · Reviewer_8Saz · 2026-03-18

**Summary Of Contributions:**

## Summary
The paper claims that existing works on preference optimization tends to assign uniform weighting across preference pairs, and claims that more (less) informative pair should force the policy to learn more (less) aggressively.
To this end, the paper proposes to adaptively attributing each preference pair based on learned explicit reward margin.
Specifically the paper describes a weight function that puts more (less) emphasis on larger (smaller) reward margin---the weight function is designed such that the gradients are stable.
The paper provides a theoretical analyses demonstrating that (1) under certain assumptions the optimization with the weight function amplifies low-margin pairs, (2) the deviation between using learned and optimal explicit reward as the objective is bounded, and (3) the gradient and hessian of the loss with respect to the policy is bounded.
Finally, the paper provides empirical results on (1) the proposed method performing better on a synthesized IMDB preference dataset, (2) the proposed method is robust to some hyperparameters, and (3) ablation on amplification and regularization.

## Strengths
- The paper motivation is clear and the paper is generally well written.
- The theoretical result demonstrates that the weighting function does amplify the low-margin pairs with optimal reward.
- The result indicates that amplification on specific pairs is critical for good performance.

## Comments
- There are certain design choices that seems arbitrary and written in a way so that the method is limited to these choices
  - Since the algorithm estimates the reward model anyway, why does the algorithm not use the policy optimization in the RLHF formulation and weigh the particular pairs based on the margin?
  - The function $c(h_\phi)$ can be anything so long it satisfies the condition that it outputs $> 1$ for low margins and $< 1$ for high margins right? Specifically in the proof it seems like so long as $h_\phi$ satisfies the regularity conditions we get the desired theoretical results.
  - In the experimental setup, the reward value is bounded to be $[-3, 3]$, is this choice swept? Secondly it can be argued that this reward range is tied to the solution method, thus each algorithm should have its own transformation.
  - Regarding the data generation, how are high/medium/low quality determined? The terminology is confusing as one could argue that SFT model generation is of low quality. Can the paper clarify?
- Regarding the theoretical analysis
  - Assumption 4.3 suggests that each $\phi$ is bounded by a different $B$ which I suspect is not the case, it should be written as "there exists a $B$ such that for all $\phi \in \Phi$, ..."
  - Assumption 4.5 mentions a constant $L_\theta$ that doesn't seem to be dependent on the specific values of $\theta$---perhaps drop the subscript?
  - Regarding the result on theorem 4.6, while there is some discussions about the two terms, there is no discussion on whether the second term is vacuous.

**Audience:**

Yes

**Audience Explanation:**

The paper focuses on enhancing preference-based learning, which is one of the core methodologies nowadays for fine-tuning LLM. While the current scope of the paper is somewhat limited, there could be impact on the development of improving data efficiency based on adaptive margins.

**Broader Impact Concerns:**

N/A.

**Claims And Evidence:**

No

**Claims Explanation:**

In addition to the comments in the summary, I have the following concerns:
- The results for both proposition 4.1 and proposition 4.2 seem to apply for the whole dataset, regardless of the pair margin. However, the main paper presents them as if they are for specific margins based on $\tau$. The core issue here is that there is no discussion about the samples excluded in the propositions. In proposition 4.1, the policy will have learned a reduced signal on the high margin pairs. Likewise for proposition 4.2, even for low-margin data the gradient is greater than 0---as a consequence I don't completely understand the utility of proposition 4.2
- On page 9, first paragraph, the paper indicates that the policy cannot assign a probability of exactly 0 or 1---I agree with this statement (due to the sigmoid function) but I fail to understand why this justifies assumption 4.5.
- The paper mentions that $\beta = 0.1$ across all methods---this seems to be one of the most critical hyperparameter that for some reason is not swept. Each algorithm should have their best $\beta$ compared.
- Finally, the paper spends some effort indicating the losses are bounded with optimal/learned reward, and that optimization is predictable, e.g. bounded gradient and hessian. However there is no empirical results that corroborate these findings.

**Requested Changes:**

Based on the concerns above, I request the following:
- More discussion for the theoretical results, specifically on the data excluded from the proposition/theorem statements.
- Each algorithm should have $\beta$ swept and compare each algorithm based on the best $\beta$.
- Empirical results showing that the learning is stable, secondly that the learned model is close to the ground-truth reward model that is accessible as described in the experiment section.

---

> ### Author Response · Authors · 2026-03-18
>
> Thank you for the detailed and constructive feedback. We are currently running new experiments to address your concerns and updating the manuscript. In the meantime, we wanted to answer your theoretical questions and share our experimental plan to ensure it meets your expectations.
>
> **Planned experiments and methodology**
>
> * Regarding why we do not use full RLHF policy optimization: We avoid methods like PPO because they require expensive online sampling and are prone to instability. Our goal with MADPO is to maintain the offline simplicity and training stability of DPO while fixing its uniform weighting issues.
> * We agree with your concerns about the synthetic IMDB dataset and the reward transformation. To address this, we are currently running new experiments on the real openai/summarize_from_feedback dataset using a larger model. In this new setup, we define the high, medium, and low quality tiers based on annotator confidence instead of our previous heuristic.
> * As requested, we are also running a full sweep of the $\beta$ hyperparameter for all baseline algorithms so we can compare their best performances fairly.
> * To support our theoretical claims about stable optimization, the revised paper will include new empirical plots. These will show that the empirical losses, gradients, and the distance between the learned and ground-truth reward models remain well-behaved during training. We will upload these results as soon as the runs are complete.
>
> **Theoretical clarifications**
>
> * You are correct that the mathematical results in Propositions 4.1 and 4.2 apply to the entire dataset, not just the specific margins. We presented them this way purely for explanation purposes, to highlight how the method amplifies or dampens the signal where it matters most. We will clarify in the text that these properties hold globally across the dataset.
> * Regarding the weight function $w(h)$: In theory, any function with $c>1$ for low margins and $c<1$ for high margins would satisfy the conditions. However, our specific piecewise design is necessary in practice to prevent the gradients from exploding when dealing with large negative margins.
> * Thank you for catching the phrasing error in Assumption 4.3. We will correct the manuscript to formally state that there exists a constant $B>0$ such that $||\phi|| \le B$ for all $\phi$.
> * For Assumption 4.5, we used the subscript in $L_\theta$ simply to show that this constant applies to the policy network, to keep it distinct from other constants (e.g., $L$ from Theorem 4.6). We will also clarify the justification for this assumption: because the policy's output probability cannot be exactly 0 or 1, the implicit margin $h_\theta$ remains bounded. This ensures that the $\log(\sigma(\cdot))$ term does not diverge.
> * For Theorem 4.6, we will add a discussion explaining that the second term is non-vacuous when the data is sparse. It serves to bound instability in under-sampled reward directions.
>
> We will post another update with the empirical results and the revised PDF once the new experiments are finished.

---

> ### Author Response · Authors · 2026-04-20
> **Author Response and Summary of Revisions**
>
> We appreciate the reviewer’s thorough assessment of our theoretical framework and experimental design. To address these concerns, we have significantly updated the manuscript, moving from a synthetic sentiment task to a real-world summarization benchmark and refining our mathematical notation.
>
> 1. Design Choices: The reviewer asked why we do not use the full RLHF formulation if we are already estimating a reward model. MADPO is designed to preserve the primary advantage of DPO—bypassing Online RL (PPO) and its associated sampling costs and training instability. Training an offline reward model is significantly cheaper than the iterative sampling required by PPO. MADPO offers a stable, instance-level solution for static data that remains resource-efficient.
>
> 2. Flexibility of the Weight Function $c(h_\phi)$: The reviewer noted that the coefficient function could theoretically be any function satisfying certain output conditions. We agree and have updated the manuscript to note that our theoretical results (Propositions 4.1 and 4.2) hold for any generalized function $c$ satisfying the required boundary conditions. We instantiated $c(h_\phi)$ using the continuous function in Equation 4 for our empirical evaluation because it is flexible enough to cover a wide range of functional forms while allowing for a smooth transition between amplification and dampening. Also, our specific piecewise design is necessary in practice to prevent the gradients from exploding when dealing with large negative margins.
>
> 3. Experimental Setup: The reviewer raised concerns regarding the arbitrary reward range and the determination of data quality tiers in our initial IMDB setup. We have replaced the synthetic IMDB experiments with the Reddit TL;DR Summarization benchmark. Instead of heuristic quality tiers, we now utilize real human-annotated preference data with recorded confidence scores. This provides a verifiable and realistic distribution of preference clarity.
>
> 4. Theoretical Clarifications and Notation: We thank the reviewer for the precise feedback on our assumptions:
> * Assumption 4.3: We have corrected the phrasing to state that there exists a universal constant $B>0$ such that $||\phi||\le B$ for all parameters in the space.
> * Assumption 4.5: We have retained the subscript $L_\theta$ to clearly distinguish this constant as a property of the policy network. We have added a justification: because the policy cannot assign a probability of exactly 0 or 1, the implicit margin remains bounded, which ensures the log-likelihood term does not diverge.
> * Propositions 4.1 and 4.2: We have rewritten Propositions 4.1 and 4.2 to apply globally across the dataset. We now explicitly specify how each proposition supports the distinct amplification (low-margin) and regularization (high-margin) components of MADPO.
> * Theorem 4.6: we have added a discussion explaining that the second term is non-vacuous when the data is sparse. It serves to bound instability in under-sampled reward directions.
>
> 5. Hyperparameter Fairness ($\beta$ Sweep): The reviewer correctly identified that the base temperature $\beta$ is a critical hyperparameter. We no longer use a fixed $\beta$. For all methods (MADPO and all baselines), the base temperature is now tuned over a discrete range of $\{0.01, 0.05, 0.1\}$. Our results now compare each algorithm based on its best-performing hyperparameter configuration found on a separate validation subset.
>
> 6. Empirical Stability Results: As requested, we have added empirical evidence to corroborate our theoretical findings regarding stability. We included a stress test using 30% adversarial (inverted) labels. Figure 3 shows that MADPO maintains bounded, stable gradients perfectly aligned with Vanilla DPO, confirming the stability guarantees of Proposition 4.7.

---

### Review · Reviewer_gCWH · 2026-03-30

**Summary Of Contributions:**

This paper proposes MADPO, a method that adaptively re-weights the DPO loss at the instance level based on the reward margin estimated by a separately trained reward model. The paper provides theoretical analysis showing the method achieves its intended amplification/regularization behavior, is Lipschitz continuous and robust to reward estimation errors, and preserves the optimization geometry of DPO. Experiments on a synthetic sentiment generation task using IMDB show gains over DPO.

**Audience:**

Yes

**Audience Explanation:**

Preference optimization is an important and widely studied topic in LLM research, and the findings would be of broad interest if they were supported by rigorous experimental evidence.

**Claims And Evidence:**

No

**Claims Explanation:**

1. All experiments use a single 270M-parameter model (Gemma-3-270M) on a single dataset (IMDB) for a single task (sentiment generation). This is far below the standard expected for preference optimization papers.
2. All preference data is generated via an oracle RoBERTa model with Gumbel noise. While the authors acknowledge this limitation, the gap between synthetic and real human preference data is significant. There is no evidence the method works on real-world preference data.
3. Although SimPO (Meng et al., 2024) and α-DPO (Wu et al., 2024a) are mentioned, they are not included in the empirical comparisons. The paper should also clarify its positioning relative to online RL methods such as GRPO (Shao et al., 2024).

**Requested Changes:**

1. Scale the experiments to include larger models on standard benchmarks using real preference data.
2. Add comparisons against more baseline methods, as noted in Concern 3 above.

---

> ### Author Response · Authors · 2026-04-01
>
> Thank you for reviewing our paper and highlighting the importance of evaluating our method on actual preference data. We have put together a plan to address your concerns and would appreciate your feedback.
>
> 1. We agree that validating on real human preference data is important. To address this, we are replacing the synthetic IMDB experiments with the standard TL;DR summarization benchmark, which uses actual human preferences. We will also scale up our experiments by evaluating MADPO on Qwen 2.5 7B using QLoRA. This directly bridges the gap between synthetic and real world data using a modern large language model.
>
> 2. While you requested scaling to multiple larger models across different benchmarks, we want to align our experimental scope with the official TMLR acceptance criteria. TMLR evaluates papers based on technical correctness and whether the empirical evidence supports the specific claims made, rather than requiring state of the art results across exhaustive benchmarks. We believe that a rigorous evaluation on a 7B model using a real world dataset, combined with our theoretical proofs of stability, provides sufficient evidence for our claims.
>
> 3. You suggested including SimPO and $\alpha$ DPO as empirical baselines. However, these methods tackle a fundamentally different problem by focusing on removing the need for an explicit reference policy. MADPO is designed to solve the uniform weighting issue within the standard reference based DPO framework. Comparing them empirically does not add direct information to our specific claims. Regarding GRPO and online RL methods, we will update the text to explicitly contrast our offline method with them. Online methods require active generation and reward scoring during the optimization loop, which is a completely different paradigm.

---

> ### Author Response · Authors · 2026-04-20
> **Author Response and Summary of Revisions**
>
> We thank the reviewer for highlighting the limited empirical scope of the original submission. We agree that validation on real human preference data and at a more realistic model scale is important for supporting our claims. To address this concern, we have substantially revised the experimental section.
>
> 1. Scalability and real-world evidence. We have replaced the original Gemma-270M synthetic sentiment experiments with experiments on Qwen2.5-7B-Instruct fine-tuned with LoRA on the Reddit TL;DR summarization benchmark, using human preference data. In the revised manuscript, Section 5 and Figure 1 now report results in this setting, where MADPO consistently outperforms vanilla DPO, IPO, and β-DPO across decoding temperatures. We believe this updated experimental setup provides much stronger evidence for the paper’s main claims in a realistic offline preference optimization setting.
>
> * We agree that empirical evidence should be sufficient for the claims being made. While some preference-optimization papers pursue broader scale or benchmark coverage, our contribution here is a technically grounded instance-level adaptation method, supported by both revised real-data experiments and theoretical analysis. After replacing the original synthetic setup with a 7B-scale experiment on human preference data, we believe the paper now provides evidence that is appropriately matched to its claims.
>
> 2. Positioning relative to SimPO and α-DPO. We appreciate the suggestion to clarify our relationship to SimPO and α-DPO. In the revised manuscript, we expanded the related-work discussion to make this distinction clearer. These methods are relevant adjacent approaches, but they primarily focus on reference-free preference optimization, whereas MADPO is proposed as an instance-level margin-adaptive modification of standard reference-based DPO. For this reason, our empirical comparisons focus on methods that address the same core setting and failure mode, namely DPO, IPO, and β-DPO. We have clarified this positioning in the manuscript.
>
> 3. Contrast with online RL methods such as GRPO. We have also revised the discussion section to better distinguish MADPO from online RL approaches such as GRPO. MADPO is an offline, sampling-free preference optimization method, whereas GRPO-style methods require on-policy generation and reward evaluation during training. We do not present MADPO as a substitute for all online RL methods, but rather as a practical and stable offline alternative for settings where one wants fine-grained control over preference optimization without online rollout costs.

---

### Author Response · Authors · 2026-04-21
**General Response and Summary of Revisions**

We thank the reviewers and the Action Editor for their constructive feedback. We have substantially revised the manuscript to address concerns regarding empirical scale, data realism, and theoretical rigor.

## Key Revisions include:
* Scalability & Real-World Data: We have replaced the synthetic sentiment experiments with a 7B-scale evaluation. We now validate MADPO using Qwen 2.5 (7B) fine-tuned with LoRA on the Reddit TL;DR Summarization benchmark, utilizing actual human preference data.

* Theoretical Rigor: We corrected Assumptions 4.3. We added a discussion on Theorem 4.6 to demonstrate its stability in sparse data regimes, ensuring the results remain non-vacuous in under-sampled reward directions. Furthermore, Propositions 4.1 and 4.2 were reformulated as global guarantees.

* Empirical Stability: We added a stress test with 30% adversarial labels and included Gradient Stability plots (Figure 3) to corroborrate our theoretical proofs regarding bounded gradients and optimization stability.

* Fairness in Comparison: We performed a full hyperparameter sweep for the base temperature ($\beta$) across all baseline methods (DPO, IPO, $\beta$-DPO) to ensure a fair comparison at each algorithm’s peak performance.

## Guide to the Revised Manuscript:
For the convenience of the reviewers, we have used blue text to highlight critical changes throughout the document. However, please note the following exceptions made for readability:
* Section 5 (Experiments): This section has been entirely rewritten to reflect the new 7B-scale results and is not highlighted in blue.

* Appendix B: This section has also been added to provide updated experimental details and is not highlighted.

* Organizational Shift: The Related Work section has been moved from Appendix B to Appendix C.We believe these updates provide evidence that is now appropriately matched to our claims, focusing on technical correctness and theoretical-empirical alignment in a realistic alignment setting.

---

### Decision · Action_Editor_8WLj · 2026-05-30

**Recommendation:** Reject

**Audience:**

Yes

**Audience Explanation:**

Preference optimization and alignment of large language models are important topics, and the paper addresses a problem that is very relevant to the community.

**Claims And Evidence:**

No

**Claims Explanation:**

The proposed approach relies heavily on reward model-estimated margins, but the paper provides limited analysis of its sensitivity to reward model quality and calibration. It remains unclear whether the observed gains stem from the proposed adaptive weighting mechanism or simply from additional information introduced through the reward model. Even after revision, one reviewer continued to raise this concern after revision. Without such evidence, the main hypothesis underlying the method remains insufficiently validated. In addition, several reproducibility and evaluation details remain unclear, such as the final hyperparameter settings, the treatment of draws in win-rate calculations, and the precise definition of gradients used in the empirical stability analysis. Therefore, I believe a major revision is required before acceptance.

**Resubmission Of Major Revision:**

The authors may consider submitting a major revision at a later time.